# Recent Advances in BODIPY Compounds: Synthetic Methods, Optical and Nonlinear Optical Properties, and Their Medical Applications

**DOI:** 10.3390/molecules27061877

**Published:** 2022-03-14

**Authors:** Prabhuodeyara M. Gurubasavaraj, Vinodkumar P. Sajjan, Blanca M. Muñoz-Flores, Víctor M. Jiménez Pérez, Narayan S. Hosmane

**Affiliations:** 1Department of Chemistry, Rani Channamma University, Belagavi 591156, India; vinodkumarsjjn@gmail.com; 2Facultad de Ciencias Químicas, Universidad Autónoma de Nuevo León, San Nicolás de los Garza 66451, Nuevo León, Mexico; blanca.munozfl@uanl.edu.mx; 3Department of Chemistry and Biochemistry, Northern Illinois University, DeKalb, IL 60115, USA

**Keywords:** organoboron, synthesis fluorescence, nonlinear optical, imaging

## Abstract

Organoboron compounds are attracting immense research interest due to their wide range of applications. Particularly, low-coordinate organoboron complexes are receiving more attention due to their improbable optical and nonlinear optical properties, which makes them better candidates for medical applications. In this review, we summarize the various synthetic methods including multicomponent reactions, microwave-assisted and traditional pathways of organoboron complexes, and their optical and nonlinear properties. This review also includes the usage of organoboron complexes in various fields including biomedical applications.

## 1. N-B-N Environment in Organoboron Compounds

The chemistry of organoboron compounds is one of the most multifaceted research areas among heteroatom-substituted organic molecules. These organoboron compounds have vast applications in numerous fields including biomedical and nuclear chemistry [1,2,3,4]. There is a dramatic rise in the research on the applications of boronic acid and their derivatives [5,6,7,8,9]. Boron is an interesting molecule with an electron-deficient character and is also more electropositive than carbon. This rudimentary property of boron has been completely utilized in synthesizing various organoboron compounds and exploring their applications in organic synthesis [10,11,12,13,14,15].

The report by Treibs and Kreuzer on BODIPY derivatives have opened up an exciting and useful field in chemistry. Since then, BODIPY chemistry has grown immensely and reports have poured in for the different ways of synthesis of BODIPY compounds and their applications. These applications include their uses in laser dyes, protein tags, and metal sensors. Fluorescent compounds have seen the limelight as these are most importantly studied by various research communities in multidisciplinary areas. Among all the other fluorescent compounds, boron-containing compounds are of utmost interest these days as they have significant and thrilling applications in various fields as active media of tunable lasers; development of photoelectronic devices, fluorescent probes, and chemical sensors; or monitoring the physicochemical characteristics of the surrounding ambiences. They also have optical features, as these compounds show better photo-stability, robust fluorescence intensity, high quantum yields, and small Stokes shift [16]. Even though these systems are known for intrinsic potential applications, their photophysical properties are highly focused so as to design new dyes with specific properties. This can be performed by changing the molecular structure of the chromophore (substituent effect) and the environmental conditions (solvent effect, incorporation in rigid solid materials, etc.) [17].

In this review, we summarize the various synthetic methods including multicomponent reactions, microwave-assisted and traditional pathways of organoboron complexes, and their optical and nonlinear properties. This review also includes the usage of organoboron complexes in various fields including biomedical applications. 

## 2. Organoboron Compounds Having NBN Framework

In recent times, there has been increasingly immense research interest in BODIPY (boron-dipyrromethene) compounds containing distinct substituent groups (with heteroelements in meso and other positions) based on NBN ligand core [18]. This is due to their attractive properties as they are tunable for fluorescence emission in 500–700 nm regions with high fluorescent quantum yield in various solutions and good photostability [19,20,21,22]. These fluorescent compounds have found profound applications as tracers in fluorescence microscopy in fluorescence immunoassay and in flow cytometric analysis, along with a series of other useful applications [23,24,25,26,27,28].

In this review, we summarize a series of new meso-polyarylamine-BODIPY hybrids of the general structure (A) Figure 1, which were synthesized by two different modified methods.

(A)Liebeskind–Srogl coupling: Cross-coupling of thiomethyl BODIPYs with arylaminoboronic acids [29].(B)Liebeskind–Srogl and Suzuki coupling: By a two-step sequence, a reaction to prepare meso-bromoaryl BODIPYs followed by coupling of these Bromine-containing BODIPYs with arylaminoboronic acids [30,31,32,33,34,35].

Several of these derivatives exhibited emission in the near-infrared region. BODIPY derivatives of 2-thienyl and 2,6-bisthienyl displayed intense absorption and a large Stokes shift in contrast with the typical BODIPY.

Based on DFT calculations [36,37,38], it was proposed that the large Stokes shifts of **3**, **4**, and **5** (Figure 2) are due to the remarkable geometry relaxation upon photoexcitation and its substantial effect on the energy levels of molecular orbitals. For the dyes with small Stokes shifts, much smaller geometry relaxations were found [39,40,41].

Several research groups reported [42] the detailed synthesis and reactions of aza-boron-dipyrromethene (Aza-BODIPY) compounds (Figure 3) containing methoxy and hydroxyl groups. The study on linear absorption spectra for phenolate forms of aza-BODIPY containing hydroxyl group exhibited drastic changes and showed new bands for phenolate groups in the region below 500 nm and above 700 nm in THF solutions. In addition, no fluorescence signals were observed with 600 nm excitation for phenolate forms. Moreover, these hydroxyl group (HABDP)-containing azo-BODIPY compounds revealed two photon absorption properties at 1200–1450 nm spectral range [43].

Reports [44] on the properties of aza-BODIPY containing triphenylamine, 4-ethynyl-N,N-dimethylaniline, and methoxy moieties (Figure 4) such as substitution and charge transfer on linear and nonlinear optical absorption (especially two-photon absorption) were investigated by ultrafast pump–probe spectroscopy technique. It was observed that aza-BODIPY compounds with good electron-donating moieties (triphenylamine and 4-ethynyl-N,N-dimethylaniline moieties) have charge transfer from electron-donating parts of the molecules to aza-BODIPY core. The two-photon absorption cross sections increase with the electron-donating strength.

Additionally, 1,8-naphthyridine–BF_2_ complexes **15**–**18** were synthesized (Figure 5); these are known for their good fluorescence properties. These complexes contain one N atom less between naphthyridine moieties that have strong emissions in the solid state (Figure 1) [45]. Both naphthyridine and pyridine units in their structure are ligated to the BF_2_ core as monodentate ligands, and the two aromatic units are nearly coplanar with dihedral angles of 4.91 and 2.968 for the B-and N-form crystals, respectively. SEM and TEM images of 17 showed that it consists of tangled nanowires of width about 30 nm and lengths varying from several hundred nanometers to several micrometers (Figure 2).

Further, some research groups reported a new dye, BF_2_-rigidified anilido-pyridine boron difluoride (Figure 3), which show large Stokes shifts and high photostability [46]. These are air- as well as moisture-stable and do not undergo photodegradation even upon exposure to continuous radiation. This photostability makes the dye more efficient when compared with BODIPY and many other dyes. Their efficacy as probes for biological membranes was demonstrated using a liposome model.

Besides, other aryl and hetaryl moieties in BODIPY compounds are widely reported. Three two-photon active boradiazaindacene derivatives 2,6-di-phenylacetylenyl-1,3,5,7-tetramethyl-8-phenyl-4,4-difluoroboradiazaindacene (Figure 4, **24a**), 2,6-di-9-ethyl-9H-carbazole-3-ethynyl-1,3,5,7-tetramethyl-8-phenyl-4,4-difluoroboradiazaindacene (Figure 4, **24b**) and 2,6-di-4-N,N-diphenyl-phenylacetyleny l-1,3,5,7-tetramethyl-8-phenyl-4,4-difluoroboradiazaindacene (Figure 4, **24c**) in THF solutions were studied by using femto-second laser spectroscopic techniques [47]. The two-photon fluorescence imaging experiment on these compounds exhibit good cell permeability, nontoxicity, and excellent two-photon fluorescence properties. Structurally rigid BODIPY having spirofluorene moieties [48] (Figure 5) were reported that exhibit intense bathochromic fluorescence. These rigid structures give high quantum yield of photoluminescence and decreased nonradiative decay of excited states. DFT calculations indicated that spiro-conjugation leads to delocalization of the *π*-system of BODIPY derivatives over the fluorene moieties as well as the BODIPY core. Moreover, symmetric BODIPY dyads (Figure 6) have chromophores at the meso position through phenylene bridge or direct linkage [49]. In excited state, these molecules will undergo symmetry-breaking through ICT state. Due to differences in degree of rotational freedom, these dyads will show different behavior of the ICT state. Whereas dyad **25** undergoes rapid nonradiative decay to the ground state, the more hindered dyad **26** has a long-lived ICT state with moderate-to-high fluorescence quantum efficiency. The excited state properties of these dyads could prove useful in facilitating charge separation in photovoltaic devices.

Introduction of radiotracers in BODIPY compounds is also known [50]. The rapid nucleophilic [^18^F]-radiolabeling of a BODIPY dye in aqueous solutions is reported (Figure 7). This radiolabeled dye was found to be stable in vivo and used as a dual modality imaging agent. Besides several applications of BODIPY compounds, we elucidate that the BODIPY-based fluorescent probe **31** can be used for the selective detection (Figure 8) of tyrosinase (a copper-containing enzyme catalyzing the hydroxylation of phenol derivatives, such as tyrosine or tyramine, which is widespread in plants and animal tissues) activity in buffered aqueous solution [51], is suitable for screening potential inhibitors of tyrosinase, as well as for bioimaging intracellular tyrosinase activity in living cells.

Some NBN-environment-based organoboron compounds have also been reported with a wide variety of applications. Thus, optically active organoboron aminoquinolate-based coordination polymers bearing the chiral side chain derived from L-alanine were synthesized (Figure 6) and their optical behavior was studied by UV–Vis and photoluminescence spectroscopies [52]. The hydrogen-bonding in these polymers was found to be stable in solvents such as CHCl_3_ and DMF studies of circular dichroism (CD). Intramolecular charge transfer was observed due to the fact of solvents polarity. Further, some toxic compounds such as Triphenyl borane were studied (Figure 9). The toxicity of the alternative organotin (Ot) antifoulants TPBP (triphenylborane pyridine; Figure 9: 34) and TPBOA (triphenylborane octadecylamine; Figure 9: 35) and their degradation products on *Crassosteagigas* and *Hemicentrotuspulcherrimus* were tested [53]. Silylated-diborylene-3,4,9,10-tetraaminoperylenes (DIBOTAPs, compounds **39**–**42**) were synthesized by treating 4,9-diaminoperylenequinone-3,10-diimine (DPDI, Figure 7) with BH_3_–THF, lithiation with *n*-butyllithium, and subsequent addition of the corresponding silyl chloride (Figure 8) [54]. In all cases, the perylene backbones were found to be not completely planar. The coordination of the nitrogen donor atoms to the Lewis acidic boron atoms stabilizes the tetraaminoperylene core, while the *N*-silylation appears to suppress aggregation in solution. The latter enables the high luminescence quantum yields. The exchange of all three methyl groups with ethyl (compound **39**) or isopropyl (compound **40**) substituents resulted in a significant increase in quantum yields with values of 92% and 89%, respectively. The observed fluorescence decay is monoexponential for all dyes with typical lifetimes of 5.5–6.6 ns.

To conclude, 1D boron containing two-photon absorbing fluorophores with two boron-containing central cores (with two boron atoms)—the cyclodiborazane and the pyrazabole moieties—were reported (Figure 9) [55]. All compounds present a strong two-photon induced fluorescence and have been used in microscopy to visualize cancerous HeLa cells. High boron content should be of great interest to study the mechanism of boron neutron capture therapy by deep imaging in small animals with micrometric resolution by two-photon excited fluorescence.

## 3. Compounds Containing O-B-O Framework

A predominant number of boron compounds containing O-B-O- moiety corresponds to β-diketone derivatives, having been extensively studied for their fluorescent properties [56]. Reports on the first difluoroborate diketone compounds that are curcumin-derivatized by the NIR fluorescent probe, CRANAD-2 (Figure 10), for in vivo biological studies and provides a useful type of NIR fluorescent dye for cell, tissue, and in vivo imaging for small animals [57]. Upon interacting with aggregates, CRANAD-2 undergoes a range of changes, which include a 70-fold fluorescence intensity increase, a 90-nm blue shift (from 805 to 715 nm), and a large increase in quantum yield. After intravenous injection of this probe, 19-month-old Tg2576 mice exhibited significantly higher relative signal than that of the control mice over the same period of time (Figure 10) [58].

Surprising process-dependent and reversible mechanochromic fluorescence was discovered for the boron dodecane complex (BF_2_dbmOC_12_H_25_) (Figure 11)—a difluoroboron dibenzoylmethane dye coupled to a lipid chain [59]. A thermally annealed spin-cast film of the lipid dye on glass exhibited blue fluorescence under UV light; however, after shearing or scratching, the mechanically perturbed region turned yellow–green. The blue coloration could be rapidly recovered by thermal treatment of the film [60]. In order to test the effects of alkyl chain length on solid-state photoluminescence and reversible mechanochromic luminescence (ML) in difluoroboron β-diketonate dyes, a series of dyes, BF_2_dbmOR, with dibenzoylmethane (dbm) ligands and alkoxyl substituents (–OR) were prepared [61], where R = C*_n_*H_2*n*+1_ and *n* = 1, 2, 3, 5, 6, 12, 14, 16, 18 (Figure 12). Fluorescence spectra and lifetimes were found to be nearly identical for dyes in CH_2_Cl_2_ solution; whereas, emission maxima and lifetimes were different among the samples in the solid state as powders, thin films, or spin cast films, The recovery time generally increased with alkyl chain length, ranging from minutes (*n* = 3) to days (*n* = 18). Longer chain analogues (*n* ¼ 6, 12, 14, 16, 18) did not fully return to the original annealed emissive state even after months on quartz, though the dynamics are substrate-dependent.

The difluoroboron avobenzone complex (BF_2_AVB) (Figure 13) is commercially available and used as an ingredient in sunscreen products because of its strong absorption of UVA light (320–400 nm) was synthesized via BF_3_·OEt_2_ boronation in CH_2_Cl_2_ avobenzone [62]. Unlike BF_2_dbm(s) derivatives that typically exhibit strongly red-shifted and significantly broadened fluorescence spectra, BF_2_AVB(s) showed unexpectedly sharp emission spectra that can be tuned via the solid form, such as single crystals, dendritic solid, or spin-cast film (Figure 14). The fluorescence color was found to be dramatically altered after crushing or physically smearing BF_2_AVB crystals or upon scratching or rubbing annealed film samples.

The discovery of an exceptional group of boron-containing compounds, the borolithochromes, causing the distinct pink coloration of well-preserved specimens of the Jurassic red alga Solenoporajurassica, (Figure 15) was reported in [63]. The borolithochromes are characterized as complicated spiroborates (boric acid esters) with two phenolic moieties as boron ligands, representing a unique class of fossil organic pigments. Although the borolithochromes originated from a fossil red alga, no analogy with hitherto known present-day red algal pigments has been found. The occurrence of the borolithochromes or their possible digenetic products in the fossil record may provide additional information on the classification and phylogeny of fossil calcareous algae. Finally, boron measurements at subcellular scale are essential in boron neutron capture therapy (BNCT) of cancer as the nuclear localization of boron-10 atoms can enhance the effectiveness of killing individual tumor cells. Thus, the secondary ion mass spectrometry (SIMS)-based imaging technique of ion microscopy was used [64] to quantitatively image the boron from two BNCT agents, clinically using *p*-boronophenylalanine (BPA) and 3-carboranylalkylthymidine(N4) (Figure 16) in mitotic metaphase and interphase human glioblastoma T98Gcells. N4 belongs to a class of experimental BNCT agents, designated 3-carboranylthymidine analogues (3CTAs), which presumably accumulate selectively in cancer cells due to a process referred to as kinase-mediated trapping (KMT). The cells were exposed to BPA for 1 h and N4 for 2 h. The BPA-treated interphase cells revealed significantly lower concentrations of boron in the perinuclear mitochondria-rich cytoplasmic region compared with the remaining cytoplasm and the nucleus, which were not significantly different from each other. In contrast, the BPA-treated metaphase cells revealed a significantly lower concentration of boron than cytoplasm in their chromosomes. In addition, the cytoplasm of metaphase cells contained significantly less boron than the cytoplasm of interphase cells.

## 4. O-B-N Boronates

A series of boronates **56a**–**56b** were synthesized by the single step reaction of 2,4-pentanedione, aminophenol, and phenylboronic acid in good yield (Figure 11) [65]. The compounds crystallized in centrosymmetric space groups are useful for the growth of organic crystals with luminescent and nonlinear optical properties. The crystals were used to prepare aqueous colloidal nanocrystals that exhibited superior fluorescence properties to those of the boronates when dissolved in organic solvents. This image shows a photograph of the luminescence observed from the colloidal solution (Figure 17); for comparison, this figure also shows the absence of fluorescence from a chloroform solution of **56b** with the same molar concentration as the colloidal solution of nanocrystals.

N,O-containing bororganic compounds are also common [66]. Thus, four diboron-contained ladder-type π-conjugated compounds **57**–**60** (Figure 12) were designed and synthesized [67]. Compounds **57** and **58** possess high thermal stabilities, moderate solid-state fluorescence quantum yields, as well as stable redox properties, indicating that they are possible candidates for emitters and charge-transporting materials in electroluminescent devices. The third-order nonlinear optical characterization of a boronate (**58**)—prepared from the reaction of diphenylboronic acid and the bidentate ligand (**57**)—derived from 4-dimethylaminocinnamaldehyde (Figure 18), was performed by third-harmonic generation (THG) at the infrared wavelength of 1550 nm [68]. The results showed that the N→B coordinative bond facilitates the polarization of the electronic π-system, a situation that optimizes the third-order nonlinear optical (NLO) response. In addition, three boron complexes (**65a**–**65c**) were prepared by the reaction of bidentate ligands (**66a**–**66c**) and diphenylboronic acid (Figure 13) [69]. Compounds **58a** and **58c** were found to have a nonplanar conformation for the main p-backbone, acquired after boron complexation; for compound **58b**, the planar conformation is preserved.

A series of fluorescent boron systems **67a**−**67c** and **68a**−**68d** based on nitrogen (NNN) or nitrogen and oxygen (ONO)-containing tridentate ligands were prepared (Figure 14) [70]. They showed large Stokes shifts (mostly above 3200 cm^−1^) and quantum yields in solution and in the solid state up to 40%. Introducing a long alkyl chain with a phenyl spacer at this axial position enables the self-assembly of the boron compound **68d** to form a fluorescent vesicle, which is able to encapsulate small molecules such as sulforhodamine. Additionally, boron compound **68d** was found to serve as a dye for cell imaging since it has the capability of binding to the nuclear membrane cells. A boron complex bearing a pyrene ligand (CPB) as fluorophore was synthesized (Figure 15) and introduced as the first example of a binuclear boron complex inorganic light-emitting diode [71]. Complex CPB exhibited strong red-light emission in the solid state. In the polymer light-emitting diodes fabricated with the CPB complex blended with PVK, red emission could be achieved easily by tuning the weight concentration of CPB.

A series of boron ketoiminate derivatives that exhibited clear aggregation-induced emission (AIE) characteristics (in THF, FPL =0.01; in the solid state, FPL = 0.30–0.76) were prepared by the reactions of 1,3-enaminoketone derivatives with boron trifluoride–diethyl etherate (Figure 16) [72,73,74,75,76,77,78,79,80,81,82,83]. The boron ketoiminate units can be applied as a new building block of various AIE-active materials. The reaction of 8-hydroxyquinoline (HQ) with B(C_6_F_5_)_3_ led [84,85,86,87,88] to the formation of the zwitterionic compound (C_6_F_5_)_3_BQH (Figure 17). On the basis of these and other results, it was shown that fluorination of the phenyl rings results in a stabilization of both the HOMO and LUMO levels; therefore, the effect on the absorption and emission maxima in the UV–Vis and PL spectra, respectively, is only minimal. However, the difference in stability and volatility between fluorinated and unfluorinated luminescent boron compounds may have an effect on their solid-state properties and their performance in OLED devices.

Among benzoxazole and benzothiazole derivatives, two π-conjugated organoboron complexes **80** and **81** (Figure 19) with rigid seven-ring fused core structures bridged by boron atoms and highly efficient red (632 nm) and deep red (670 nm) solid-state fluorescence were constructed [89,90,91,92,93,94] (Figure 18) and qualified as potential nondoped red emitters accompanied by excellent electron-transport ability. The two side phenyl groups coordinated to each boron atom effectively keep the luminescent units apart. As a result, these red fluorophores are brightly fluorescent in the solid state (fluorescence quantum yields: 0.30 for **80** and 0.41 for **81**). Their emission spectra are shown (Figure 20). 2-(20-Hydroxyphenyl)benzoxazole (HBO) and 2-(20-hydroxyphenyl)benzothiazole(HBT) reacted with triphenylborane produced two rigid p-conjugated fluorescent cores: **82**(BPh_2_(BOZ), BOZ 2-(benzo[d]oxazol-2-yl)phenol); **83** (BPh_2_(BTZ), BTZ 2-(benzo[d]thiazol-2-yl)phenol) [95,96,97,98,99]. Simple modification of these frameworks (Figure 19) allowed the synthesis of strongly fluorescent materials **84** (BPh_2_(para-Cz-BTZ), Cz 9H-carbazol-9-yl), **85** (BPh_2_(para-NPh_2_-BOZ), NPh_2_diphenylamino), **86**, and **87** (BPh_2_(para-NMe_2_-BTZ), NMe_2_dimethylamino). Organic light-emitting diodes employing these boron complexes as emitters not only kept the full-color tunable emission feature but also showed high electroluminescent performance.

Complexation of boron trifluoride by a series of electron donor/acceptor substituted 2-(20-hydroxy phenyl) benzoxazole (HBO) derivatives resulted in luminescent B(III) complexes **106**–**114** (Figure 19) with an emission wavelength ranging from 385 to 425 nm in dichloromethane or toluene [100]. Depending on the nature of the substituents present on the core of the starting substituted 2-aminophenol I and 2-hydroxybenzaldehyde II, two different routes were chosen. Route A involved refluxing I and II together in ethanol to obtain the cyclic carbinolamines, which precipitated from the reaction mixture (Figure 20). After collection, these compounds were oxidized with slight excess of 2,3-dichloro-5,6-dicyano-1,4-benzoquinone (DDQ). The second one-pot route B involves the oxidation-sensitive substituents, such as diethylamino groups, in presence of phenylboronic acid and requires potassium cyanide to promote benzoxazole cyclization. The synthesized dyes can be connected to other photoactive subunits such as BODIPY or Boranil cores to afford sophisticated molecular cassettes. In addition, four diboron-bridged, π-conjugated ladder molecules **115** were designed (Figure 21) and synthesized (Figure 21) [101]. It was revealed that the bulky phenyl substituents on boron centers efficiently prevented π stacking of the luminescent ladder unit. The construction of diboron-containing ladder-type skeletons endowed these materials with good thermal stability, high fluorescence quantum yields, and strong electron affinity. Simple EL devices fabricated using complexes **116** and **117** as both emitter- and electron-transporting materials exhibited the highest brightness and efficiency among boron-containing materials reported so far. Finally, fluorescent homopolymers and amphiphilic block copolymers were prepared by reversible addition–fragmentation chain transfer (RAFT) polymerization of two styryl-type organoboron monomers (Figure 22) [102,103,104,105]. Block copolymers featuring a relatively long PEO segment formed stable micellar solutions in water with luminescence characteristics similar to those of the respective (water-insoluble) homopolymers, suggesting potential applications as nanosized fluorophores in biological environments.

Among other interesting O-B-N-containing organoboron compounds are the so-called “push–pull” type of molecules [106,107,108]. These compounds derive from the well-known stilbene backbone, to which an arylboron (ArB) fragment has been added. This family of readily available macrocyclic boron compounds has recently attracted some interest from various perspectives in analytical and supramolecular chemistry. Thus, a series of eighteen such molecules were obtained by self-assembly of salicylidenimino phenols and various phenylboronic acids [109,110,111,112,113,114]. Such compounds can be prepared according to the reactions where a monomeric boronate and an oxobridged chiral dimer were obtained by reaction of the ligand derived from 4-diethylaminosalicylaldehyde with (R)-(R)-phenylglycinol and phenyl boronic acid or boric acid (Table 1) [115,116,117,118,119,120,121]. The existence of the N–B coordination bond was established by ^11^BNMR, which showed the characteristic signal at 4.0, 2.1, and 6.1 ppm for **125**, **122a**, and **123b**, respectively (Figure 23). Electric-field-induced second-harmonic measurements of the nonlinear optical response revealed that the nature of the phenyl-boron moieties has a modest influence on the molecular hyperpolarizabilities.

Solvent effects on the spectroscopic and photophysical properties of tris{[*p*-(N,N-dimethylamino)phenylethynyl]-duryl}borane (TMAB) and tris[(phenylethynyl)duryl]borane (TPhB) (Figure 22) were studied [122,123,124,125,126,127,128,129,130,131,132,133,134,135]. Both TMAB and TPhB exhibited broad and structureless absorption and fluorescence bands attributed to the charge transfer (CT) transition between the *π*-orbital of the aryl group (*π*(aryl)) and the vacant *p*-orbital on the boron atom (p(B)): *π*(aryl)-p(B) CT.

## 5. Conclusions

In this review, we summarized various synthetic methods of BODIPY-based organoboron compounds with different frameworks. We also summarized the various optical and nonlinear properties of these compounds along with their applications. BODIPY compounds based on NBN network were synthesized by Liebeskind–Srogl; Liebeskind–Srogl and Suzuki coupling showed intense absorption and large Stokes, unlike the typical BODIPY due to the geometry relaxation. The DFT calculations supported the geometrical relaxation upon photoexcitation and its remarkable effect on the energy levels of molecular orbitals. Moreover, boron compounds containing O-B-O- upon interaction with aggregates increase the fluorescence by 70-fold to 90 nm blue shift and significantly increase in quantum yield. New compounds such as the boron dodecane complex (BF_2_dbmOC_12_H_25_) have emerged with dependent and reversible mechanochromic fluorescence.

A series of fluorescent boron systems based on nitrogen (NNN) or nitrogen and oxygen (ONO)-containing tridentate ligands were reported. They showed large Stokes shifts and quantum yields in solution and in the solid state. Introducing a long alkyl chain with a phenyl spacer at this axial position enables the self-assembly of the boron compound to form a fluorescent vesicle, which is able to encapsulate small molecules such as sulforhodamine. Furthermore, few boron compounds were found to serve as a dye for cell imaging since it has the capability of binding to the nuclear membrane cells. Moreover, a boron complex bearing a pyrene ligand (CPB) as fluorophore was synthesized and introduced as the first example of a binuclear boron complex inorganic light-emitting diode. Complex CPB exhibited strong red-light emission in the solid state. In the polymer light-emitting diodes fabricated with the CPB complex blended with PVK, red emission could be achieved easily by tuning the weight concentration of CPB.

Finally, fluorescent homopolymers and amphiphilic block copolymers were prepared by reversible addition–fragmentation chain transfer (RAFT) polymerization of two styryl-type organoboron monomers. Block copolymers featuring a relatively long PEO segment formed stable micellar solutions in water with luminescence characteristics similar to those of the respective homopolymers, suggesting potential applications as nanosized fluorophores in biological environments.

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
