# Peer review of "Recent Advances in BODIPY Compounds: Synthetic Methods, Optical and Nonlinear Optical Properties, and Their Medical Applications"

_molecules, 2022, doi:10.3390/molecules27061877_

Round 1

Reviewer 1 Report

Gurubasavaraj, Pérez and Hosmane et al summarized the recent development of tetracoordinate organoboron compounds. They systematically reviewed the synthetic methods and applications of optical and nonlinear properties. I recommend to publish it in Molecules after minor revision. However, there are a lot of issues to be concerns as listed below.

Main text:

1) P. 2 Organoboron compounds having NBN framework, line 2 “ These complexes contain one N atom less between naphthyridine moieties, that have strong emissions both in solution and in the solid state (Figure 1) ”
Figure 1 has only solid-state emission spectra, which do not reflect the strong emissions of both in solution and in the solid states; besides, the molecules 10-14 does not correspond to the molecules in scheme 5, which is misleading.

Cross-reference

1) P. 6 Organoboron compounds having NBN framework, line 1 “~(Figure 4, 21a), ~(Figure 4, 21b), ~(Figure 4, 21c)”
Figure 4 clearly shows molecule 24a – 24c.

2) P. 16 O-B-N Boronates, line 1 “A series of boronates 1a-1b were synthesized by the single step reaction of 2,4-pentanedione, aminophenol and phenylboronic acid in good yield (Scheme 11)”
The scheme 11 clearly shows molecule 56a – 56b. And What are the corresponding R groups for the 56c and 56d molecules in scheme 11?

3) P. 17 Figure 17 title,” Photoluminescence from boronate 1b in a chloroform solution (left in the picture) and 323 from a colloidal solution of nanocrystals (right in the picture). Fluorescence was obtained under 324 excitation at 370 nm.”
The boronate 1b corresponds to which molecule in scheme11?

4) P. 17 O-B-N Boronates, last 4 lines “In addition, three boron complexes (58a–58c) were prepared by the reaction of bidentate ligands (57a–57c) and diphenylborinic acid (Scheme 12) [69]. The compounds 58a and 58c were found to have a non-planar conformation for the main p-backbone is acquired after boron complexation; for compound 58b the planar conformation is preserved.”
The molecule numbers in this should match those in scheme 12. Also, due to the duplication of scheme12, the subsequent numbering is all wrong. Please check carefully and make corrections.

5) P. 21 O-B-N Boronates, line 2” Among benzoxazole and benzthiazole derivatives, two -conjugated organoboron complexes 1 and 2 (Figure 19) ~” and line 8” apart. As a result, these red fluorophores are brightly fluorescent in the solid state (fluorescence quantum yields:0.30 for 1 and 0.41 for 2). ”
The complexes 1 and 2 in this should match those in Figure 19.

6) P. 22 Figure 20 title” Emission spectra (a) and photographs under UV light (365nm) (b) of 82–85 in CH2Cl2 and the solid state.”
How 82-85 corresponds to molecules 1-4 in the diagram should be explained.

Format of the annotation of the literature:
1) P. 2 Reference 5
2) P. 5 Reference 14
3) P.6 Reference 16 and 17
4) P. 7 Reference 16
5) P .23 Reference 45

Reviewer 2 Report

This review is focused on recent advances in the synthesis and investigation of BODIPY derivatives. Currently, these compounds continue to attract the attention of many scientific groups owing to their remarkable fluorescent properties and chemical robustness. The review is conveniently structured, and for each compound class, the information regarding the synthetic methods, emission properties, and biomedicine-related applications is discussed. In general, the authors managed to maintain the necessary balance between detailed descriptions and generalizations. The manuscript itself is well written and well-illustrated. In my opinion, this review will be very useful both for specialists working in the field of BODIPY chemistry and for the general community dealing with these compounds. Therefore, I recommend the publication of this review in Molecules after minor revision. The following concerns should be addressed in the revised manuscript: 
1. Almost all the compounds discussed contain no direct B–C bond, and therefore, they, strictly speaking, can’t be considered as “organoboron” compounds. 
2. The title of the review does not accurately reflect its content. In fact, the review is devoted to the BODIPY family, but not to “Organoboron Compounds” overall. Thus, the title should be appropriately corrected. 
